# Multi-Task Reinforcement Learning with Language-Encoded Gated Policy Networks

## Abstract

Multi-task reinforcement learning often relies on task metadata—such as brief natural-language descriptions—to guide behavior across diverse objectives. We present Lexical Policy Networks (LEXPOL), a language-conditioned mixture-of-policies architecture for multi-task RL. LEXPOL encodes task metadata with a text encoder and uses a learned gating module to select or blend among multiple sub-policies, enabling end-to-end training across tasks. On MetaWorld benchmarks, LEXPOL matches or exceeds strong multi-task baselines in success rate and sample efficiency, without task-specific retraining. To analyze the mechanism, we further study settings with fixed expert policies obtained independently of the gate and show that the learned language gate composes these experts to produce behaviors appropriate to novel task descriptions and unseen task combinations. These results indicate that natural-language metadata can effectively index and recombine reusable skills within a single policy.

## 1 Introduction

Multi-task reinforcement learning (MTRL) aims to train a single agent that solves many tasks and reuses skills across them. Recent MTRL approaches condition behavior on task metadata—often short natural-language descriptions—to improve generalization and sample efficiency.

'Context-Aware Representations', or CARE, introduced by Sodhani et al. (2021) exemplifies this trend by leveraging natural language metadata about the different tasks to generate representations. In CARE, context, or metadata, embeddings are used to gate over the outputs of multiple state-embeddings networks to generate a single combined state representation. The state representation is concatenated with the context embedding and ingested by a single policy to generate the resulting actions for the agent.

While CARE became a benchmark for multi-task reinforcement learning, it doesn't entirely reflect how humans tend to reason about their environment. Humans often master multiple smaller sub-skills and then combine them in different capacities (Graybiel, 1998; Jin & Costa, 2010) to solve new tasks—the amount of different skills, or policies, required to solve a goal often change between different tasks and even in different states of the same task. This quantization into smaller skills must, therefore, serve as a motivation for building agents that can perform well in multi-task reinforcement learning.

We introduce Lexical Policy Networks (LEXPOL) as a novel algorithm for multi-task reinforcement learning. LEXPOL uses language-encoded gated policy optimization—the same single state is provided to multiple policies and the final action is picked by gating over the outputs of the different policies using natural-language embeddings of the task context. Since all individual policies receive the same state representation, one can use LEXPOL to combine multiple single-task skills to perform larger multi-task learning. The entire algorithm can be learned end-to-end.

We evaluate LEXPOL on MetaWorld using the evaluation protocol of CARE, reporting success rate and sample efficiency. LEXPOL matches or exceeds strong multi-task baselines under end-to-end training. We also study a frozen-experts setting in which sub-policies are expert single-task policies trained independently (e.g., via single-task RL or scripted controllers). With only the gating module learned, LEXPOL composes

these experts using language to produce appropriate behavior for novel task descriptions and unseen task combinations.

Both LEXPOL and CARE work by disentangling the complexity of multi-task reinforcement learning into smaller learnable pieces. CARE divides the state into object-specific and state-specific representations with a universal policy, while LEXPOL divides the tasks into fundamental skills that are then combined into a universal action. As a final step, we propose preliminary experiments combining LEXPOL and CARE to encapsulate their respective skill and state decompositions into a single hybrid framework.

Our primary contributions are:

1. We introduce *Lexical Policy Networks* (LEXPOL), a language-conditioned mixture-of-policies architecture that uses natural-language task metadata to gate over the actions of multiple sub-policies, enabling modular skill reuse in multi-task RL.

2. We empirically evaluate LEXPOL on the MetaWorld domain and show that it matches or exceeds strong baselines in success rate and sample efficiency.

3. We qualitatively evaluate LEXPOL using frozen-experts and compare it to end-to-end learning to demonstrate the modular skills learned generalize to novel task descriptions.

4. We combine LEXPOL with Context-Aware Representations (CARE) into a hybrid algorithm that jointly factorizes *behaviors* (policy) and *representations* (state), yielding additional gains in higher-data regimes.

## 2 Background

A **Markov Decision Process (MDP)** $(\mathcal{S}, \mathcal{A}, \mathcal{T}, \mathcal{R}, \gamma)$ (Sutton & Barto, 2018; Bellman, 1957) consists of a set of states $\mathcal{S}$, a set of actions $\mathcal{A}$, a transition function $\mathcal{T}(s, a, s') \coloneqq P(\mathcal{S}_{t+1} = s' | \mathcal{S}_t = s, \mathcal{A}_t = a)$, a reward function $\mathcal{R} : \mathcal{S} \times \mathcal{A} \to \mathbb{R}$ and a reward *discount factor* $\gamma \in [0, 1)$. Since we consider a multi-task setting, each goal has its own reward function $\mathcal{R}_g$ and discount factors $\gamma_g$. A Markovian *policy* $\pi : \mathcal{S} \times \mathcal{A} \to [0, 1] \coloneqq P(\mathcal{A}_t = a | \mathcal{S}_t = s)$ is a probability distribution of all the actions conditioned over the states.

The agent's goal is to construct a policy that maximises the state value function $V_\pi(s)$ and action-value function $Q_\pi(s, a)$ defined as $V_\pi(s) = \mathbb{E}_\pi \left[ \sum_{t=0}^{\infty} \gamma^t \mathcal{R}_{t+1} \mid s_0 = s \right]$ and $Q_\pi(s, a) = \mathbb{E}_\pi \left[ \sum_{t=0}^{\infty} \gamma^t \mathcal{R}_{t+1} \mid s_0 = s, a_0 = a \right]$. Solving for the optimal action-value function allows one to deduce a greedy optimal policy. For discrete MDPs there is at least one optimal policy that is greedy with respect to its action-value function.

A **Contextual Markov Decision Process (CMDP)** $(\mathcal{C}, \mathcal{S}, \mathcal{A}, \mathcal{M})$ (Hallak et al., 2015) augments the MDP with the context space $\mathcal{C}$ and mapping function $\mathcal{M}(c) = \{\mathcal{R}^c, \mathcal{T}^c\}$ that maps context $c \in \mathcal{C}$ to rewards and transitions. CMDPs are applicable to multi-task reinforcement learning since each task can be defined as an MDP with all tasks sharing the same state space $\mathcal{S}$. For each task in the family of MDPs, the agent only has access to a subspace of the state space $\mathcal{S}^c$ consisting of all relevant objects and states. The state space $\mathcal{S}^c$ and $\mathcal{R}^c$ can differ for MDPs (tasks), but the object-specific dynamics remain consistent across tasks (even though the MDP dynamics $\mathcal{T}^c$ differ due to different $\mathcal{S}^c$).

A common starting point for MTRL is to model each task as an MDP and optimize a single policy over a distribution of tasks. In our work, tasks come with auxiliary information—natural-language descriptions—that is observed at the beginning of each episode and fixed for its duration. Treating this information as a *context* variable leads naturally to a CMDP formulation, which lets us precisely distinguish the per-episode task identity (context) from the within-episode Markov dynamics.

Like CARE (Sodhani et al., 2021), we focus on the setting where $\mathcal{S}^c$ is a strict subset of the dimensions in $\mathcal{S}$. This leads to the definition of the Block-Contextual MDP.

A **Block-Contextual Markov Decision Process (BC-MDP)** $(\mathcal{C}, \mathcal{S}, \mathcal{A}, \mathcal{M}')$ (Zhang et al., 2020; Du et al., 2019) augments the CMDP by defining the function $\mathcal{M}'$ that maps a context $c \in \mathcal{C}$ to MDP parameters and observation space $\mathcal{M}(c) = \{\mathcal{R}^c, \mathcal{T}^c, \mathcal{S}^c\}$.

The MetaWorld domain over 50 distinct robotics manipulation tasks is an ideal BC-MDP candidate since all tasks share the same state dimensionality but have different semantic meaning depending on task. For instance, the same state space could reflect a goal-state in one task and the object's location in another. Therefore, this translates into a family of MDPs with different rewards and states (semantic meaning), but with the same state-dimensionality.

In any single-task, actor-critic or policy gradient algorithms like Soft Actor-Critic (SAC, Haarnoja et al. (2018)) and Proximal Policy Optimization (PPO, Schulman et al. (2017)) would result in the ideal performance and hence result in an upper bound. However, multi-task variations of these algorithms struggle with the states having different semantic meaning depending on the task. The purpose of LEXPOL, and CARE, is to use the task context for gating over multiple networks that learn smaller pieces of information which when combined help construct a solution to different tasks. In CARE, this translates to combining different state networks to build and represent relevant state information about the task. In LEXPOL, this translates to learning smaller behaviors and skills that can be combined to produce longer-term behaviors that solve solve multiple different tasks. LEXPOL leverages the fact that, in the single-task setting, SAC and PPO learn optimal performance.

## 3 Lexical Policy Networks

Lexical Policy Networks (LEXPOL) tackle complexity in multi-task reinforcement learning by factoring the tasks into fundamental and reusable skills common across MDPs. The primary source of complexity in learning multiple-tasks arises from the reusability across the dynamics in BC-MDPs—the same section of the state space can represent different objects depending on the task (object-representation) or different skills to be performed (skill-representation). For instance, the tasks "Close the door" and "Open the window" in a BC-MDP will have the same state dimensionality, but the same dimensions in one task could represent "close" and "door" while representing "open" and "window" in another. Architectures that do not utilize the context, such as any multi-task variations of policy optimization algorithm, fails to learn the task. However, if provided with single-tasks, policy optimization algorithms can achieve perfect performance.

Factoring the state spaces of the different BC-MDPs allows us to leverage that on single-tasks, policy optimization algorithm achieve benchmark performance. Having a collection of optimal policies on factored skills poses the advantage that they can be combined in different capacities to form more complex skills. We will indeed see that in our experiments that learning an attention over optimal single-task skills results in solve multiple longer-horizon tasks of higher complexity.

LEXPOL use natural language embeddings to leverage the task context for learning a gating over the different skills. Language serves as a favorable medium for context as metadata is often available or easily constructed for different tasks. The entire architecture ranging from the gating to policy parameters is learned end-to-end, but we will also note in Section 4.2 that it is possible to just learn the gating and context embeddings over already learned optimal policies.

### 3.1 Architecture

The architecture for lexical policy networks is divided into three distinct parts:

1. **Context Encoder:** The raw natural language instruction is encoded into a fixed dimension using the a Pre-Trained Language model and an optional multi-layer perceptron. The output is a single $n$-dimensional encoded context $z_{context} \in \mathbb{R}^n$. We use BERT (Devlin et al., 2019) as our pre-trained language model.

2. **Mixture of Policies:** A mixture of policies is used to learn the factorized skills that then combine into solving multiple longer horizon task. We learn $k$ different policies, each of which produces an $m$-dimensional action $a_i \in \mathbb{R}^m$. Any policy optimization algorithm can be used for learning the policy parameters; we use Soft Actor-Critic Haarnoja et al. (2018) as our policy optimization algorithm.

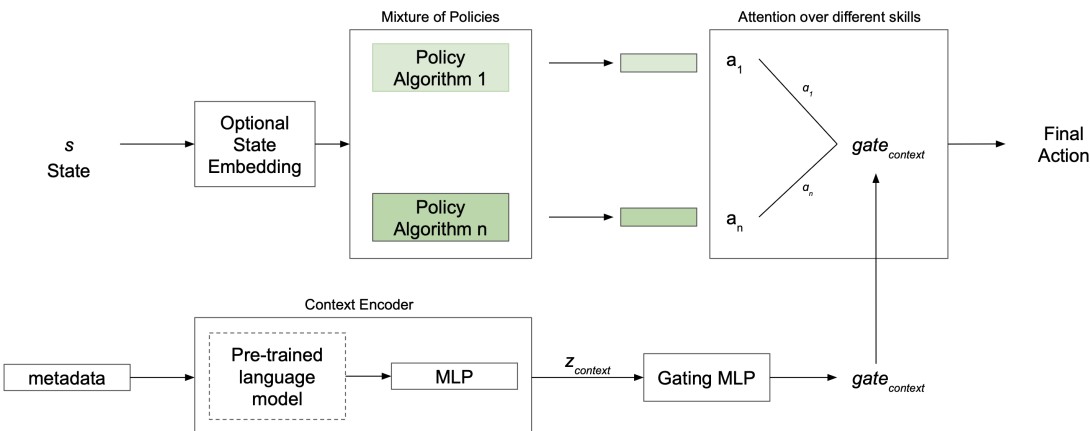

Figure 1: LEXPOL Architecture. There are three primary components: 1) A Context Encoder, 2) Mixture of Policies, 3) Gating MLP. Natural language metadata is used to encode a context while a mixture of policies, representing smaller skills, generate a series of actions. The Gating MLP is used to generate a soft attention over the policy outputs using the encoded context, resulting in a final action. Each policy represents a smaller skill that when combined with other skills can used to solve multiple longer horizon tasks.

All policies use the same state representation for inputs. We will note in Section 5 that the object-specific context representations can be combined with the factorized policies of LEXPOL to further enable multi-task learning.

3. **Gating:** A gating multi-layer perceptron $\mathbb{R}^n \to \mathbb{R}^k$ transform $z_{context}$ into the context gating weights $gate_{context}$. These weights are used to gate over the outputs of the $k$ different policies to produce a final action $a \in \mathbb{R}^m$ that is used to interact with the environment.

Figure 1 plots the architecture for LEXPOL with the three different components highlighted. The entire algorithm is learned end-to-end; however, we will note in our experiments in Section 4.2 that it is possible to freeze the policy weights and learn a gating over prior optimized policies.

## 3.2 Algorithm

The flow of LEXPOL is defined in Algorithm 1.

The task context (metadata) is passed to the pre-trained language model $C$ to obtain $z_{context} \in \mathbb{R}^n$. The state representation $s$ is passed to all $k$ policies $P_i, \quad \forall i \in \{1, \ldots, k\}$ to generate its respective actions $a_i$. The language context embedding is converted to a softmax gating embedding $g_{context}$ by passing it to the gating MLP $G$. The final action is the dot product of the attention weights and the policy outputs. The policy loss is used to update all the parameters as all networks directly result in the generation of the final action.

The hyperparameter here is the number of policies $k$, which also controls the output size of the gating MLP. We use Soft Actor-Critic (Haarnoja et al., 2018) as our policy optimization algorithm. While the algorithm describes a method of learning the entire task end-to-end, we will see in our experiments that it is possible to freeze the policy parameters and just learn the gating. This leaves it to the user to learn single-task modular policies that would be effective when combined.

---

**Algorithm 1** Lexical Policy Networks

---

**Input**: State Representation $s$
**Require**: Pre-Trained Language Model $C$
**Require**: Mixture of $k$-Policies $P_i$
**Require**: Gating MLP $G$
**Output**: Final Action $a$

1: Let $t = 0$.
2: **for** timestep $t = 1..N$ **do**
3:    **for** each task $T_m$ **do**
4:       $z_{context} = C(metadata)$
5:       $a_i = P_i(s), \quad \forall i \in \{1, \ldots, k\}$
6:       $\bar{z}_{context}^m = stopgrad(z_{context}^m)$
7:       $g_{context}^m = G(\bar{z}_{context}^m)$
8:       $\alpha_i = softmax(g_{context}^m)$
9:       $a_m = \sum_{i \in \{1, \ldots, k\}} a_i^m \times \alpha_i$ (OR $\bar{\alpha}.\bar{a}_i$)
10:      Update parameters $P, G$
11:    **end for**
12: **end for**

---

### 3.3 Comparison with State Disentanglement

We draw our closest comparison with Context-Aware Representations (CARE, Sodhani et al. (2021)) which performs gating over the states instead of actions with a similar motivation of breaking down multi-task complexity into modular representations. CARE decomposes the task representations, such as object representations, by breaking the state representation into a mixture of state-encoders with a single universal policy. LEXPOL decomposes the task into modular skills by using a mixture of policies with a single state. LEXPOL is further motivated by the fact that humans often don't learn a one universal skill, but instead learn a magnitude of smaller behaviors that collectively solve multiple longer horizon tasks.

We will see in Section 5 that it is possible to combine LEXPOL and CARE to utilize both, modular skills and modular representations, to further enhance multi-task learning.

## 4 Experiments

Our experiments focus on the task performance of the end-to-end learning of LEXPOL as compared to benchmark baselines, along with an interpretation of LEXPOL on tasks where the policy is pre-trained and the parameters are frozen. The efficacy of using task metadata in multi-task learning was already demonstrated by Sodhani et al. (2021), and hence we shall not cover it.

### 4.1 LEXPOL Performance

#### 4.1.1 Method

We conduct experiments on the MetaWorld domain (Yu et al., 2019)—a popular multi-task and meta-learning robotics domain consisting of 50 different benchmark tasks. It offers two versions of tasks, **MT10** (multi-task 10) and **MT50** (multi-task 50), that simulate multi-task learning over 10 and 50 robotics domains respectively. It comprises 50 diverse robotic manipulation tasks, each challenging agents with different object interactions—ranging from pushing and pulling to opening and closing objects like drawers, doors, and cups. Although every task shares a common state and action space, the semantic interpretation of these states varies across tasks, thereby making it an ideal candidate for a BC-MDP.

We compare LEXPOL with its closest method Context-Aware Representations (CARE, Sodhani et al. (2021)), which we described as a benchmark method that uses natural language to gate over state representations. We also compare our method with Attenion-based Mixture-of-Experts (AMESAC, Cheng

et al. (2023)) which uses a backbone network to extract domain knowledge with a task-conditioned attention mechanism without using task priors such as metadata. We compare with Soft Modularization (Yang et al. (2020)) that performs routing in a shared-policy network to learn policies for different tasks. We compare with SAC + FiLM (Perez et al., 2018), which is a general-purpose conditioning method used to condition the CARE encoder on context. Finally, we compare against Multi-Task SAC as a simple extension of a single-task RL algorithm to multi-task settings. We use single-task SAC (Haarnoja et al., 2018) as the upper-bound of measuring performance.

We use the same evaluation strategy used by Sodhani et al. (2021) in CARE in order to ensure uniformity across the previous benchmark method. The agent is evaluated at regular intervals by conducting 5 trials in each test environment, with the mean success rate across these trials representing performance in the respective environment. These environment-specific success rates are subsequently averaged to yield a mean success rate for each evaluation interval. By repeating this evaluation process at regular intervals throughout training, we construct a time series representing the progression of mean success rates. Since the agent is trained with multiple random seeds (ten in our experiments), we obtain ten distinct time series, each corresponding to a different seed. These ten series are further averaged to calculate the overall mean success rate across seeds. Ultimately, we report the best mean success rate observed across all time-series as the agent's performance metric. This evaluation method is necessary since MetaWorld returns binary success signals at the end of its episodes.

### 4.1.2 Results

Results are reported for the MT10 and MT50 setups of tasks to include a varied difficulty of the problem. Further, the results are reported at two intervals of 100000 timesteps and 2 Million timesteps. We find that LEXPOL outperforms other methods in all sets of tasks.

Table 1 plots the results for the **MT10** setup after 100k and 2 million timesteps. In both settings, LEXPOL achieves the best performance. LEXPOL also demonstrates better sample efficiency as seen by its performance in the low-sample setting of 100k timesteps. Interestingly, using a Mixture-of-Encoders underperforms at the low-sample setting.

Table 1: Evaluation Performance Comparison of LEXPOL with previous benchmark methods on the **MT10** test environments after **100k timesteps** and **2 million timesteps** of training for each environment. The results are averaged. LEXPOL matches or outperforms the benchmark methods, including the two primary baselines of CARE and Mixture-of-Encoders (MoE). Single-Task SAC (one SAC per task) is used as the upper-bound of performance. Statistical significance denoted by *

| Agent | Success @ 100k (mean ± stderr) | Success @ 2M (mean ± stderr) |
|---|---|---|
| Multi-task SAC* (Yu et al., 2019) | $0.10 \pm 0.02$* | $0.45 \pm 0.051$* |
| Soft Modularization (Yang et al., 2020) | $0.35 \pm 0.042$ | $0.71 \pm 0.062$ |
| SAC + FiLM (Perez et al., 2018) | $0.24 \pm 0.031$* | $0.72 \pm 0.072$ |
| MOORE Hendawy et al. (2024) | $0.33 \pm 0.014$ | $0.83 \pm 0.021$ |
| SAC + CARE (Sodhani et al., 2021) | $0.35 \pm 0.038$ | $0.82 \pm 0.054$ |
| SAC + MoE (Cheng et al., 2023) | $0.29 \pm 0.056$ | $0.78 \pm 0.034$ |
| LEXPOL (**Our Algorithm**) | $\mathbf{0.39 \pm 0.052}$ | $\mathbf{0.86 \pm 0.063}$ |
| One SAC agent per task (upper bound) | | $0.93 \pm 0.051$ |

Likewise, we observe similar results on the more difficult **MT50** task. Table 2 plots the results for the MT50 task after 100k and 2 million timesteps. The performance for all methods, including the SAC upper-bound, degrades as compared to the MT10 task given the increased difficulty. We note that LEXPOL matches or outperforms the previous benchmark methods, even on the low-sample complexity setup.

Table 2: Evaluation Performance Comparison of LEXPOL with previous benchmark methods on the **MT50** test environments after **100k timesteps** and **2 million timesteps** of training for each environment. The results are averaged. LEXPOL matches or outperforms the benchmark methods, including the two primary baselines of CARE and Mixture-of-Encoders (MoE). Single-Task SAC (one SAC per task) is used as the upper-bound of performance. Statistical significance denoted by *

| Agent | Success @ 100k (mean $\pm$ stderr) | Success @ 2M (mean $\pm$ stderr) |
|---|---|---|
| Multi-task SAC (Yu et al., 2019) | $0.09 \pm 0.0072$* | $0.30 \pm 0.069$* |
| Soft Modularization (Yang et al., 2020) | $0.21 \pm 0.038$* | $0.5 \pm 0.030$* |
| SAC + FiLM (Perez et al., 2018) | $0.16 \pm 0.026$* | $0.42 \pm 0.023$* |
| MOOORE Hendawy et al. (2024) | $0.27 \pm 0.053$* | $0.60 \pm 0.067$ |
| SAC + CARE (Sodhani et al., 2021) | $0.38 \pm 0.082$ | $0.56 \pm 0.032$ |
| SAC + MoE (Cheng et al., 2023) | $0.35 \pm 0.018$* | $0.53 \pm 0.059$ |
| LEXPOL (**Our Algorithm**) | $\mathbf{0.42 \pm 0.012}$ | $\mathbf{0.64 \pm 0.057}$ |
| One SAC agent per task (upper bound) | | $0.72 \pm 0.070$ |

Statistical significance was evaluated with two-sided Welch t-tests with Bonferroni correction: LEXPOL is significantly better than baselines like Multi-task SAC, Soft Modularization and SAC + FiLM, while differences with stronger baselines like CARE/MoE are not statistically significant. Practically, LEXPOL delivers consistent learning gains under a fair tuning protocol—every method (ours and baselines) was tuned with the same search space and budget, with final configurations provided in the Appendix for reproducibility. Additional experiments and ablation studies varying the number of encoders and the language encoder are included in Appendices A.1 and A.2 respectively.

## 4.2 Pre-Trained Modular Policies

We now test the efficacy of LEXPOL when the policy parameters are fixed and only the gating and embedding MLPs can be trained.

For this experiment, we consider a continuous T-Shaped environment. There are two goals, each on opposite end: one located on the left side of the $T$ ("go to the blue goal") and the other located on the right side of the $T$ ("go to the red goal"). There are two tasks, one corresponding to each goal. The agent must navigate to the goal from any starting point. Two policies are learned to convergence using any policy optimization algorithm (SAC in our case) with one policy corresponding to each task. The parameters for each policy are then frozen.

LEXPOL is now trained on a new composite task "go to the red goal, then the blue goal" using the frozen policies—the actions are now only generated using a combination of the converged policies. The reward structure is designed to give a reward when the red goal is reached and a subsequent reward when the blue goal is reached. A penalty is incurred if the agent reaches them in the wrong order or takes too long.

We plot the percentage each policy is used in Figure 2. At each state, we plot the dominant policy with the color representing the respective red or blue policies. The intensity of the color corresponds to the percentage of policy dominance, with darker shades implying a more decisive policy selection by LEXPOL.

LEXPOL learns how to use the two frozen policies to solve the combined longer horizon task, thereby demonstrating that the usage of factorized skills that enable multi-task reinforcement learning when combined using natural language context.

Interestingly, we observe similar results when LEXPOL is trained end-to-end without pre-training the skills. The two learned skills with end-to-end LEXPOL training are the two pre-trained modular policies. This further confirms the motivation of LEXPOL for decomposing complex long-horizon multi-task setups into fundamental skills.

LEXPOL with Fixed Policy Weights

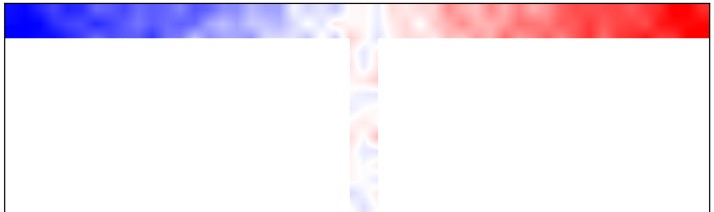

LEXPOL

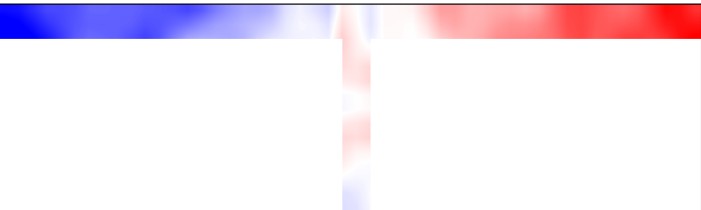

Figure 2: A comparison of LEXPOL with pre-trained frozen policies (top image) and end-to-end trained policies (bottom image). Two goals are selected, go left (*go to the blue goal*) and go right (*go to the red goal*). In the top image with pre-trained policies, two separate policies are trained corresponding to each task, their parameters are frozen, and then LEXPOL is trained on a new task (*go to the red goal then the blue goal*). In the bottom image, LEXPOL is trained end-to-end, and then trained on the new task with the policies frozen. In both cases, we see the usage of factorized skills that enable multi-task reinforcement learning on longer-horizons using natural language context. Interestingly, it is also inferred by the similarity that the end-to-end training results in the two pre-trained skills.

## 5 Combining State and Policy Context Awareness

### 5.1 Architecture

We have introduced LEXPOL that factorizes the task into a set of fundamental skills and then uses natural language context to generate a soft attention over the skills to combine them to solve multiple longer horizon tasks. We have also discussed a closely related previous method, Context-Aware Representations (CARE), that factorizes the state into object-specific and skill-specific information and then uses natural language attention to generate a single state fed into a single universal policy.

We now propose an improvement to both methods that combines both methods to leverage the best of both, state and policy context awareness—not only factorizing the state into its core components, but also using

a selection of factorized modular skills for solving longer tasks. This new method, **LEXPOL + CARE**, builds on the same architecture presented in Figure 1. A Mixture of State-Encoders is added along with its own Gating MLP and Context Encoder. The single state generated using the soft-attention over the Mixture of State-Encoders is used to generate the *State Representation* in Figure 1.

It should be noted that the same Context Encoder can be used for the Mixture of Policies and Mixture of State-Encoders if there are no parameters with it other than the pre-trained language model. If there is an additional network, as seen in Figure 1, then separate Context Encoders must be used.

### 5.2 Experiments

We now conduct experiments in the MetaWorld environment to compare LEXPOL + CARE with other methods using the same evaluation and experiment methodology as in the previous section.

Table 3: Evaluation Performance Comparison of LEXPOL + CARE with previous benchmark methods on the **MT10** test environments after **100k timesteps** and **2 million timesteps** of training for each environment. LEXPOL + CARE outperforms other methods over 2 million timesteps, but not after 100k timesteps given the increased learning difficulty with network size.

| Agent | Success @ 100k (mean ± stderr) | Success @ 2M (mean ± stderr) |
|---|---|---|
| SAC + CARE (Sodhani et al., 2021) | $0.35 \pm 0.038$ | $0.82 \pm 0.054$ |
| SAC + MoE (Cheng et al., 2023) | $0.29 \pm 0.056$ | $0.78 \pm 0.034$ |
| LEXPOL (**Our Algorithm** ) | $\mathbf{0.39 \pm 0.052}$ | $0.86 \pm 0.063$ |
| LEXPOL + CARE (**Our Algorithm**) | $0.30 \pm 0.029$ | $\mathbf{0.90 \pm 0.075}$ |
| One SAC agent per task (upper bound) | | $0.93 \pm 0.051$ |

Table 4: Evaluation Performance Comparison of LEXPOL + CARE with previous benchmark methods on the **MT50** test environments after **100k timesteps** and **2 million timesteps** of training for each environment. LEXPOL + CARE outperforms other methods over 2 million timesteps, but not after 100k timesteps given the increased learning difficulty with network size.

| Agent | Success @ 100k (mean ± stderr) | Success @ 2M (mean ± stderr) |
|---|---|---|
| SAC + CARE (Sodhani et al., 2021) | $0.38 \pm 0.082$ | $0.56 \pm 0.032$ |
| SAC + MoE (Cheng et al., 2023) | $0.35 \pm 0.018$ | $0.53 \pm 0.059$ |
| LEXPOL (**Our Algorithm**) | $\mathbf{0.42 \pm 0.012}$ | $0.64 \pm 0.057$ |
| LEXPOL + CARE (**Our Algorithm**) | $0.33 \pm 0.081$ | $\mathbf{0.68 \pm 0.049}$ |
| One SAC agent per task (upper bound) | | $0.72 \pm 0.070$ |

Table 3 plots the results for the MT10 setup after 2 million and 100k timesteps respectively, while Table 4 plots the results for the MT50 setup after 2 million and 100k timesteps respectively. LEXPOL + CARE outperforms the benchmark baselines in the MT10 and MT50 setups after 2 million timesteps, indicating the efficacy of both state and policy disengtanglement in multi-task reinforcement learning. However, LEXPOL + CARE does not outperform other methods in the low sample complexity setting of 100k timesteps since the increased number of networks is accompanied with added learning difficulty.

## 6 Related Work

There are two primary related prior works: Context-Aware Representations and Attention-based Mixture.

Context-Aware Representations (CARE, Sodhani et al. (2021)) uses natural language context to generate a soft-attention over a Mixture of State-Encoders. This generates a single combined state composed of the original state factorized into a set of object-specific state representations. The combined state is fed into a single universal policy to learn multiple reinforcement learning tasks.

Attention-based Mixtures (AMESAC, Cheng et al. (2023)) uses attention over a Mixture of Policies like we do. However, they propose using a backbone network to generate the attention by learning task embeddings directly from reinforcement learning interactions without relying on external descriptive information. Each expert network takes features from the shared backbone network and produces expert-specific outputs, specifically the Key and Value vectors. While AMESAC proposes itself as a self-contained method not relying on natural language metadata as context, we propose that natural language context is often widely available or easily generated for tasks—as agent continue to become more widely available, natural language will likely be a key method for humans to communicate with them.

There are other related works in multi-task reinforcement learning. PCGrad (Yu et al., 2020) performs gradient surgery by reducing interference between tasks during training. Soft Modularization (Yang et al., 2020) performs routing in a shared-policy network to learn policies for different tasks. Policy Sketches (Andreas et al., 2017) is a hierarchical reinforcement learning approach that leverages high-level, abstract "sketches" as weak supervision to guide the learning process. Distral (Teh et al., 2017), short for "Distill & Transfer Learning", is a framework that focuses on sharing knowledge among several tasks by distilling common behaviors into a single, central policy.

## 7 Discussion

We have introduced Lexical Policy Networks (LEXPOL)—an end-to-end multi-task reinforcement learning algorithm that factorizes the different tasks into a set of modular policies that are then combined together using soft-attention generated by a natural language context to solve multiple longer horizon tasks. Our method is motivated by human behavior in multi-task learning, where fundamental skills are combined together in varying capacities to solve more complex tasks. Using natural language metadata for multi-task learning is advantageous since it is widely available and can be easily generated—often representing how humans learn to combine skills using natural language thoughts and instructions.

Our experiments demonstrate LEXPOL matches or outperforms benchmark methods on multiple setups in the complex MetaWorld robotics domain. We also demonstrate the efficacy of LEXPOL when using pre-trained frozen skills, and make comparisons with the policies learned end-to-end to demonstrate the capacity of the framework in learning modular skills.

We have also introduced LEXPOL + Context-Aware Representations, a method that combines Lexical Policy Networks with another multi-task reinforcement learning benchmark. The combined method uses soft-attention over a Mixture of Policies as well as a Mixture of State-Encoders, thereby factorizing the task complexity twice into a collection of fundamental skills and object-specific representations. The hybrid LEXPOL + CARE method matches or outperforms LEXPOL and the other benchmark methods in the MetaWorld domain, yielding additional improvements over either state or policy factorization alone.

An interesting next avenue of research would be to continue exploring the combined LEXPOL + Care method and understand the interplay between the decomposed state-representations and policies.

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

# A  Appendix

## A.1  Additional Experiments

### A.1.1  MetaWorld

Like Sodhani et al. (2021), we also compare all methods at 0.5 million timesteps in order to make an accurate and fair comparison. This comparison at low sample size is inspired by Laskin et al. (2020).

These results are in Tables 5 and 6

Table 5: Evaluation Performance Comparison of LEXPOL with previous benchmark methods on the **MT10** test environments after **0.5 million timesteps** of training for each environment. The results are averaged. LEXPOL matches or outperforms the benchmark methods, including the two primary baselines of CARE and Mixture-of-Encoders (MoE). Single-Task SAC (one SAC per task) is used as the upper-bound of performance. Statistical significance denoted by *

| Agent | success (mean ± stderr) |
|---|---|
| Multi-task SAC* (Yu et al., 2019) | 0.32 ± 0.089 |
| Soft Modularization (Yang et al., 2020) | 0.63 ± 0.073 |
| SAC + FiLM (Perez et al., 2018) | 0.59 ± 0.017 |
| SAC + CARE (Sodhani et al., 2021) | 0.66 ± 0.034 |
| SAC + MoE (Cheng et al., 2023) | 0.60 ± 0.076 |
| LEXPOL (**Our Algorithm**) | **0.69 ± 0.093** |

Table 6: Evaluation Performance Comparison of LEXPOL with previous benchmark methods on the **MT50** test environments after **0.5 million timesteps** of training for each environment. The results are averaged. LEXPOL matches or outperforms the benchmark methods, including the two primary baselines of CARE and Mixture-of-Encoders (MoE). Single-Task SAC (one SAC per task) is used as the upper-bound of performance. Statistical significance denoted by *

| Agent | success (mean ± stderr) |
|---|---|
| Multi-task SAC* (Yu et al., 2019) | $0.25 \pm 0.090$ |
| Soft Modularization (Yang et al., 2020) | $0.44 \pm 0.082$ |
| SAC + FiLM* (Perez et al., 2018) | $0.35 \pm 0.023$ |
| SAC + CARE (Sodhani et al., 2021) | $0.49 \pm 0.056$ |
| SAC + MoE (Cheng et al., 2023) | $0.46 \pm 0.067$ |
| LEXPOL (**Our Algorithm**) | $\mathbf{0.52 \pm 0.057}$ |

### A.1.2 Time-Series

Figure 3 plots the timeseries of the success rates for all the multi-task algorithms.

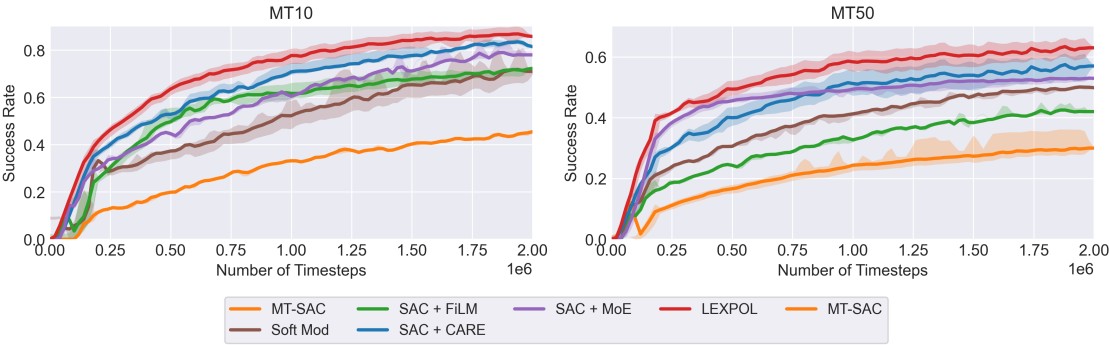

Figure 3: Success rates for the different multi-task reinforcement learning algorithms for the MT10 and MT50 domains.

### A.1.3 BabyAI/MiniGrid

BabyAI Chevalier-Boisvert et al. (2018) and Minigrid Chevalier-Boisvert et al. (2023) provide a natural domain to test since they provide grounded natural-language intructions for missions while adding a variability on the actions and descriptors in the environment, testing both action and state decomposition.

We propose a two-step curriculum over the Minigrid domain. Our task involves using two of the Minigrid environments—*Go to Object* ("go to the {color} {object type}") and *Pickup* ("pick up a {color} {object type}"). *Color* is the color of the object and can be "red", "green"", or "blue", and *type* is the type of the object and can be "ball", "box" or "key". This allows us variations in (1) the action being performed (go to vs pick), (2) the descriptive color, and (3) the type of object.

In our curriculum, we propose learning each of the two tasks and their descriptive variations separately for a set number of timesteps and then combining the tasks (for example., *Pick up blue box then Go To red ball*). In our experiments, we learn the individual tasks for 200k timesteps before switching to the complex tasks.

We report our results in Table 7. It is very interesting to note that the hybrid state and action factorizing algorithm performs best in the complex task setting that involves factorizing the actions ("Go To" vs "Pick Up") as well as the state (*color* and *object*).

Table 7: Evaluation Performance Comparison of multi-task reinforcement learning algorithms on the Minigrid environment. We aim at using tasks that introduce variations in (1) actions, (2) state descriptors (color), and (3) the objects in state (types)

| Agent | Success Rate @ 200k (Simple Multi-Task) | Success @ 500k (Complex Curriculum) |
|---|---|---|
| SAC + CARE (Sodhani et al., 2021) | $0.44 \pm 0.010$ | $0.59 \pm 0.087$ |
| LEXPOL (**Our Algorithm** ) | $\mathbf{0.42 \pm 0.027}$ | $0.60 \pm 0.073$ |
| LEXPOL + CARE (**Our Algorithm**) | $0.38 \pm 0.061$ | $\mathbf{0.65 \pm 0.033}$ |

## A.2 Ablation Studies

We conduct two ablation studies to demonstrate the effectiveness of LEXPOL under different hyperparameters along with demonstrating the lack of any significant hyperparameter tuning in our results.

### A.2.1 Varying $k$—the number of mixture-policies

We vary the number of encoders $k$ for both **MT10** and **MT50** domains. It is worth noting that for certain hyperparameters we slightly outperform the results reported in the main text of the paper (though not in a statistically significant manner), but choose not to report those since we attempt at matching the hyperparameters used in the baselines.

Table 8: Evaluation Performance Comparison of LEXPOL by varying the number of encoders on the **MT10** test environments after **2 million timesteps** of training for each environment. The results are averaged.

| Agent | success (mean $\pm$ stderr) |
|---|---|
| LEXPOL $k = 6$ (**Reported**) | $\mathbf{0.86 \pm 0.063}$ |
| LEXPOL $k = 3$ | $0.80 \pm 0.078$ |
| LEXPOL $k = 5$ | $\mathbf{0.87 \pm 0.056}$ |
| LEXPOL $k = 9$ | $0.85 \pm 0.023$ |

Table 9: Evaluation Performance Comparison of LEXPOL by varying the number of encoders on the **MT50** test environments after **2 million timesteps** of training for each environment. The results are averaged.

| Agent | success (mean $\pm$ stderr) |
|---|---|
| LEXPOL $k = 6$ (**Reported**) | $\mathbf{0.64 \pm 0.057}$ |
| LEXPOL $k = 3$ | $0.54 \pm 0.087$ |
| LEXPOL $k = 5$ | $0.62 \pm 0.047$ |
| LEXPOL $k = 9$ | $\mathbf{0.64 \pm 0.018}$ |

### A.2.2 Using alternative pre-trained language models

We used BERT to generate the encodings in the results reported in the main text. In this section, we demonstrate comparable results using alternative language models for both the **MT10** and **MT50** sets of tasks. We use the same hyperparameters as reported in all the results in the main text.

Table 10: Evaluation Performance Comparison of LEXPOL by varying the language encoder on the **MT10** test environments after **2 million timesteps** of training for each environment. The results are averaged.

| Agent | success (mean $\pm$ stderr) |
|---|---|
| LEXPOL (*BERT*) (**Reported**) | **0.86 $\pm$ 0.063** |
| LEXPOL (*RoBERTa*) | 0.83 $\pm$ 0.052 |
| LEXPOL (*DistilBERT*) | 0.78 $\pm$ 0.095 |
| LEXPOL (*ALBERT*) | 0.81 $\pm$ 0.020 |
| LEXPOL (*ELECTRA*) | 0.79 $\pm$ 0.012 |
| LEXPOL (*DeBERTa*) | 0.85 $\pm$ 0.027 |

Table 11: Evaluation Performance Comparison of LEXPOL by varying the language encoder on the **MT50** test environments after **2 million timesteps** of training for each environment. The results are averaged.

| Agent | success (mean $\pm$ stderr) |
|---|---|
| LEXPOL (*BERT*) (**Reported**) | **0.64 $\pm$ 0.057** |
| LEXPOL (*RoBERTa*) | 0.59 $\pm$ 0.081 |
| LEXPOL (*DistilBERT*) | 0.57 $\pm$ 0.086 |
| LEXPOL (*ALBERT*) | 0.62 $\pm$ 0.013 |
| LEXPOL (*ELECTRA*) | 0.65 $\pm$ 0.036 |
| LEXPOL (*DeBERTa*) | 0.63 $\pm$ 0.079 |

### A.3    Hyperparameters

We use the same hyperparameters in Sodhani et al. (2021) to make an accurate and fair comparison. These hyperparameters are listed in Tables 12 and 13.

Table 12: Hyperparameter for LEXPOL

| Hyperparameter | Values |
|---|---|
| batch size | $128 \times$ number of tasks |
| network architecture | feedforward network |
| actor/critic size | three fully connected layers with 400 units |
| non-linearity | ReLU |
| policy initialization | standard Gaussian |
| exploration parameters | run a uniform exploration policy 1500 steps |
| # of samples / # of train steps per iteration | 1 env step / 1 training step |
| policy learning rate | 3e-4 |
| Q function learning rate | 3e-4 |
| optimizer | Adam |
| policy learning rate | 3e-4 |
| beta for Adam optimizer for policy | (0.9, 0.999) |
| Q function learning rate | 3e-4 |
| beta for Adam optimizer for Q function | (0.9, 0.999) |
| discount | .99 |
| Episode length (horizon) | 150 |
| reward scale | 1.0 |

Table 13: Algorithm-specific Hyperparameter values for LEXPOL, CARE, and hybrid LEXPOL + CARE

| Hyperparameter | Hyperparameter values |
|---|---|
| task encoder size | two layer feedforward network. Hidden/output dims = 50 |
| number of encoders | 6 for MT10, 10 for MT50 (each for LEXPOL + CARE) |

Additionally, the hyperparameters used for the other methods are listed in Table 14.

Table 14: Hyperparameter values and architecture settings for (i) Soft Modularization, (ii) FiLM, (iii) SAC + MoE, and (iv) MOORE

| Method | Hyperparameter | Hyperparameter values |
|---|---|---|
| **Soft Modularization** | | |
| | task encoder size | two layer feedforward network. Hidden/output dims = 50 |
| | routing network size | 4 layers and 4 modules per layer. |
| | temperature | learned. disentangled with tasks |
| **SAC + FiLM** | | |
| | task encoder size | two layer feedforward network. Hidden/output dims = 50 |
| | temperature | learned. disentangled with tasks |
| **SAC + MoE** | | |
| | Number of experts ($M$) | MT10: 3 \| MT50: 10 |
| | task query size | 400-dim trainable task query (used by attention module) |
| | critic backbone | two-layer MLP (state-action concat $\rightarrow$ backbone features) |
| | expert network | each expert is a two-layer MLP |
| | tower network | two-layer MLP after attention mixing |
| | hidden units per layer | 400 (applies to each layer above) |
| | activation | ReLU |
| **MOORE** | | |
| *Representation Block* | | |
| | Number of Experts ($k$) | {MT10: $k = 4$, MT50: $k = 6$} |
| | Number of Linear layers | 3 |
| | Number of output units | [400, 400, 400] |
| | Activation functions | [ReLU, ReLU, Linear] |
| *Output Block* | | |
| | Number of linear layers | 1 (x number of tasks $|\mathcal{T}|$) |
| | Number of output units | [$|\mathcal{A}|$ for actor and 1 for critic] |
| | Activation functions | Linear |
| *Task Encoder* | | |
| | Number of linear layers | 1 |
| | Number of output units | Number of Experts ($k$) |
| | Use bias | False |
| | Activation function | Linear |

