# OpenReview forum: "Multi-Task Reinforcement Learning with Language-Encoded Gated Policy Networks"
_TMLR — Rejected by TMLR_

### Review · Reviewer_e4A3 · 2025-11-07

**Summary Of Contributions:**

The paper proposes Lexical Policy Networks (LEXPOL), a language-conditioned mixture-of-policies architecture for multi-task reinforcement learning. LEXPOL encodes task metadata through a pretrained language model (BERT) and uses a learned gating MLP to select or blend among multiple sub-policies, allowing end-to-end training across diverse tasks. The authors further extend this framework by integrating it with Context-Aware Representations, producing a hybrid model that leverages both state and policy context.

**Audience:**

Yes

**Audience Explanation:**

Language-conditioned multi-task reinforcement learning and modular policy architectures are active and growing areas of research. Showing that simple language-gated mixtures can match strong baselines on the MetaWorld benchmark is of interest.

**Claims And Evidence:**

Yes

**Claims Explanation:**

The paper’s main claims are supported by accurate and convincing empirical evidence. However, there are a few minor limitations that reduce clarity, although they do not undermine the overall findings.

- The algorithm pseudocode and textual explanation are inconsistent. The text states that “these weights are used to gate over the outputs of the k different policies,” yet Algorithm 1 implements $α·g_{context}$ rather than a weighted sum over policy actions.

- The paper lacks key ablations necessary to substantiate its conclusions. There is no analysis of the sensitivity to the number of experts $k$, and no examination of how the choice of text encoder affects performance.

- The paper also contains unresolved writing issues, most notably the “ADD SECTION” placeholders.

**Requested Changes:**

- Explicitly define how the final action is computed from the sub-policy outputs, and make the pseudocode consistent with the textual description. If the intended operation is a weighted sum of actions, the algorithm should clearly express $a = \sum_i \alpha_i a_i$ rather than $a = \alpha·g_{context}$.

- The empirical section should be expanded with missing ablations. The study would be stronger if it varied the number of experts $k$ to examine scalability and tested multiple text encoders, such as BERT versus smaller language models.

- Correct incomplete writing artifacts such as the “ADD SECTION” placeholders to ensure the paper’s clarity and completeness.

---

### Review · Reviewer_mjnY · 2025-11-30

**Summary Of Contributions:**

The paper proposes LEXPOL. The authors of the work propose to use the Metadata information to select the action from a mixture of policy network for Multitask RL environment. The authors used pretrained language encoder model BERT to encode the meta information. This encoded context is used to select the actions using attention like mechanism over the actions of all the policy network. The authors formulate their work to be similar to CARE and show improvements over all the baselines they studied. Further, the authors also demnostrate that their approach is able to break down the task into smaller behaviors or skills and use these to learn complex tasks. Lastly the authors demonstrate better performance when the augment their approach with CARE.
Strengths :
1. paper outlines simple approach and demonstrates superior performance.
2. Authors have added images and algorithm to clearly explain their approach.
Weakness:
See Requested changes.

**Additional Comments:**

N/A

**Audience:**

Yes

**Audience Explanation:**

I believe the topic of MTRL is interesting to the TMLR community.

**Claims And Evidence:**

No

**Claims Explanation:**

Authors have introduced their approach in the abstract and introduction. Authors should add some quantitive numbers in the abstract and introduction. Further, adding a paragraph for contributions of the work at the end of introduction would greatly enhance readabilty. Without this its hard to clearly understand the claims of the paper are. I believe that the core contribution that authors want to highligh is by introducing Mixtuer of Policy kind of action gating we fromulate the MTRL problem as composition of skills or small behaviors. This is shown by the experiment in Figure 2. However, I am not sure if this is sufficient (see feedback below).

**Requested Changes:**

1. Introduction: I believe author should improve the writing of introduction. Currently, the introduction focuses mainly on introducing what CARE does and what is difference in CARE and LEXPOL. In my opinion this should be included in the Related work section. I understand that authors assume the readers to have understanding on the topics as this is a research paper and the audience might be equiped with background knowledge. However,  I believe the paper can improve if the authors first In the introduction the problem of Multi task learning, describe challenges and how literature has tackled several aspects of this challenge, explaining the gap in the solutions (maybe they can bring in CARE to explain the gap) and then introduce LEXPOL. (To understand this work, I first had to read the CARE work and only then I was able to follow this. ).

2. Background: Similar to introduction writing of background seems abrupt to me. Authors can explain why we need BC-MDP in a coherent manner. Alternatively, authors can explain or write about MTRL problems are traditionally solved using MDP and why CMDP is required for their problem setup in the introduction. So that, when they explain the technical fromulation of these problems in Background it feels less abrupt.

3. Results : Authors can consider adding all the results in the single table, this way comparing the trends become easier. There are 2 datasets MT10 and MT50 and each datasets has results for 100k timesteps and 2 million timesteps. This is a total of 4 columns along with the column of method. This should fit in a single table. Additionally Talbes 5-8 only have 1 addition method and seems redundant to have a separate table for it.

4. Page 3 above seciton 3.1 Authors might have forgotter to replace "ADD SECTION" with section number.

5. In section 4.1.1 authors metion that they obtain a time series for sucess rates by averaging across 10 seeds and sampling sucess rates for 5 runs after every few time steps. The paper should improve if the authors could provide a plot of success rates vs time steps for several methods in a single graph.

6. The experiment in figure 2 I do not completely understand what authors are plotting here. Could authors clarify this experiment in the paper. I am confused as the complex tast is go to red and then go to blue. So agent should first go to red and after ward the go to blue policy should kick in. Hence the right part should have both blue and red policies. Am I thinking something wrong?

7. Authors at several places mention their performance is similar or superior to other methods. However, in the table, the reported number are always better. I am unsure if his is a typo or authors metion this as the results might not be statistically significant and for some seeds other methongs might outperform.

8.  Authors have based their work on CARE. All the baselines used in he work are slightly old expect the (Cheng et al 2023). Authors should consider adding newer baselines post CARE  https://scholar.google.com/scholar?as_ylo=2024&q=multi+task+reinforcement+learning+MT50&hl=en&as_sdt=0,5

9. In section 4.1.2 authors metions "every method (ours and baselines) was tuned with the same search space and budget, with final configurations provided in the Appendix for reproducibility." Authors should mention the search space and the budget. I believe hyperparameters might play a major role in the perfomrance of any of these methods.

10. Authors can compare several other parameters across the several methods for a better understanding of the work. For example, authors can compare number of trainable parameters, inference speed per timestep, training speed per time step, etc.

11. Authors have compared this methods assuming every method is trained for 2 million timesteps. Authors can also compare these approaches across iso-compute budget training or iso-wall clock time training for ablation.

12. Authors can add more ablation studies to show different aspects of their method. authors can consider looking into the related works for more experiments. Author can also include some experiments from [MTBench](https://arxiv.org/pdf/2507.23172) work.

---

### Review · Reviewer_iUhb · 2026-03-12

**Summary Of Contributions:**

To solve the multi-task learning problem and draw inspiration from humans composing sub-skills to solve complex tasks, this paper proposes a modular policy network that uses task context as metadata to produce weights for combining policies. Experiments are conducted on MetaWorld, compared with several multi-task baselines.

**Audience:**

Yes

**Audience Explanation:**

Multi-task learning is an interesting and important direction in RL, which will draw attention in the community.

**Claims And Evidence:**

No

**Claims Explanation:**

About the methodology,

The authors claimed that previous work faced such a situation: "the same section of the state space can represent different objects depending on the task (object-representation) or different skills to be performed (skill-representation)". However, it is unclear how the proposed approach handles this situation and why it fixes it.

It is unclear about the meaning of "k of policies." What's the meaning os each sub-skill? Does each task need all "k of policies" combinations?

It occurs to me that the difference between the proposed approach and Soft Modularization lies in the input information fed into the gating network. Soft Modularization uses task embedding, and this approach uses language embedding. However, from this perspective, the paper's contribution is weakened, and this should be clarified.

It is not clear if the model still works if you change the number of skills.

Would a more powerful LLM enhance the capability of the approach?

About the literature review,

The literature review in this paper is insufficient to support the claim. Quite a bit of work has been covered over the past three years.

About the results,

It is said that the architecture is compatible with any RL algorithm; however, the results only include SAC-based comparisons.

The authors should also explain why they picked the MetaWorld test and not a harder one.

Minors

Fig 2 contains less information and takes up a lot of space.

**Requested Changes:**

Please see the pros and cons part.

---

> ### Author Response · Authors · 2026-03-13
> **Rebuttal, Clarifications, and Updates to Manuscript**
>
> Dear Reviewer,
>
> Thank you for your insights and feedback! We appreciate your effort into reading our paper and recognizing its strengths such as an "interesting and important direction in RL, which will draw attention in the community."
>
> We are glad that we can address your comments by providing clarifications quite easily using the relevant parts of the paper.
>
> Addressing them in order:
> - It is possible for environments to share objects and skills. For example, we can define a set of multi-tasks such as "Pick up ball", "Push ball", "Push door", "Pick up box", etc. The two representations here would be: state-space/objects such as ball and door, or skill based such as pick up and push. Using modularity, we can take advantage of the shared object and policies. Previous works, such as context-aware reprepresentations, focus on taking the state-space and making it modular (common objects between tasks/common states between tasks). Our work focusses on making the skill modular (common skills/policies between tasks that can be weighted over to produce larger behaviors).  \
> We also combine the two into a hybrid method that combines this state-space/task and skill modularity and present results on the hybrid method, showing gains in combining skill and object based modularity. \
> To clarify further, we have tweaked the introduction.
> - "k" is a hyperparameter representing how many skills we choose to train. Not all tasks need all sub skills (a push task won't need the pull skill). This is accounted for by the gating model which picks the necessary skills needed and how much of each skill is needed for the given task/state at any time. Since it is a hyper parameter, it is possible to overfit or underfit it.  \
> We also have qualitative analysis accompanying the results to show the policy learned by each sub-skill and provide further insights.
> - Our method is quite different from soft modularization. Our closest method is CARE. Soft mod has a base policy network, with "L" modules in the network feeding into each other (output of one module is input to another). LEXPOL uses k different policies with shared inputs (instead of a single modular network in soft mod, we use a mixture of modular networks). Further, as you pointed out, we use language embeddings instead of one-hot task and state embeddings.
> - Ablation studies with different skills are in the appendix.
> - Ablation studies with different LLMs are also in the appendix. We noticed comparable results over different number of skills and different language models.
> - We have updated the literature review further upon your recommendation to place our work in relation to dissimilar but relevant multi-task RL works. Thank you! \
> Our baselines are with most relevant methods over the past two years and the most popular methods in multi-task RL.
> - *Any RL algorithm can be used to train the method since our method is agnostic to the underlying Rl algorithm. We have conducted multiple empirical experiments, qualitative analysis and ablation studies varying the environments and hyparameters. The underlying algorithm is not a hyperparameter, and hence we chose not to do an ablation over it.*
> - We picked Metaworld since it is a state-of-the-art benchmark for multi-task and meta-learning algorithms. Most popular multi-task algorithms (including many of our baselines) use metaworld as their evaluation domain. Given that it has variation over 50 unique tasks combining many different skills and objects, metaworld is a complex domain to evaluate on.\
> *We also believe that qualitiative analysis is necessary to accompany any empirical study. Hence, we conducted complementary experiments shown in Figure 2 that analyze the policies learned.* \
> *We have included additional experiments in the BabyAI/Minigrid environment as well since it is a popular domain for multi-task learning.*
>
> **Thank you for the feedback! We hope this dicussion and clarifications addresses your comments on the method and domains.**

---

### Comment · Action_Editor_Mt51 · 2026-04-29
**To authors.**

This is a quick heads up from the action editor that I am chasing the last (third) recommendation for the paper. Sorry for the delay.

---

> ### Author Response · Authors · 2026-04-29
> **Response**
>
> Thank you for the helpful update!

---

### Decision · Action_Editor_Mt51 · 2026-05-24

**Recommendation:** Reject

**Additional Comments:**

I encourage the authors to resubmit the paper when (1) all the methodological issues are fixed and (2) the authors come up with a more compelling story of what is genuinely new (or in any case what isn't new, but still interesting to the TMLR community).

**Audience:**

Yes

**Audience Explanation:**

On top of my concerns above, I am worried about novelty. The main contribution (the LEXPOL architecture, figure 1), while I cannot point to anything in literature that matches it verbatim, is based on well-established ideas and primitives (pick a policy dependent on additional information) that are combined in standard ways.

By itself, given the remit of TMLR, this would not be a reason to reject, but combined with the methodological issues above, it supports a reject decision for the current version of the paper (see my comment below about resubmitting).

**Claims And Evidence:**

No

**Claims Explanation:**

The paper is an empirical paper, which means experiments are paramount. As one of the reviewers has pointed out, there are strong remaining question marks about how the conclusions of the experimental section would change under changed hyper-parameters.

Moreover, clear claims require clear (and justifiable) definitions. The main formalism studied in the paper (block-contextual MDP) is defined in a way crucially different from the standard way of doing that present in literature [1,2], specifically without mentioning the emission function (typically denoted by q, missing from the paper). This is neither explained nor motivated. Note that this is not a nitpick or an unimportant detail - solving block-contextual MDPs is literally the main claimed contribution of the paper.

[1] Amy Zhang, Clare Lyle, Shagun Sodhani, Angelos Filos, Marta Kwiatkowska, Joelle Pineau, Yarin Gal, Doina Precup Proceedings of the 37th International Conference on Machine Learning, PMLR 119:11214-11224, 2020.

[2] Du, S., Krishnamurthy, A., Jiang, N., Agarwal, A., Dudik, M., & Langford, J. Provably efficient rl with rich observations via latent state decoding. In International Conference on Machine Learning (pp. 1665-1674). PMLR.

**Resubmission Of Major Revision:**

The authors may consider submitting a major revision at a later time.